# Providing sex and relationships education for looked-after children: a qualitative exploration of how personal and institutional factors promote or limit the experience of role ambiguity, conflict and overload among caregivers

Catherine Nixon,[1] Lawrie Elliott,[2] Marion Henderson[1]

[1]MRC/CSO Social and Public Health Sciences Unit, University of Glasgow, Glasgow, UK
[2]Department of Nursing and Community Health, Glasgow Caledonian University, Glasgow, UK

**Correspondence to**
Dr Catherine Nixon;
catherine.nixon@glasgow.ac.uk

## ABSTRACT

**Objectives** To explore how personal and institutional factors promote or limit caregivers promoting sexual health and relationships (SHR) among looked-after children (LAC). In so doing, develop existing research dominated by atheoretical accounts of the facilitators and barriers of SHR promotion in care settings.

**Design** Qualitative semistructured interview study.

**Setting** UK social services, residential children's homes and foster care.

**Participants** 22 caregivers of LAC, including 9 foster carers, 8 residential carers and 5 social workers; half of whom had received SHR training.

**Methods** In-depth interviews explored barriers/facilitators to SHR discussions, and how these shaped caregivers' experiences of discussing SHR with LAC. Data were systematically analysed using predetermined research questions and themes identified from reading transcripts. Role theory was used to explore caregivers' understanding of their role.

**Results** SHR policies clarified role expectations and increased acceptability of discussing SHR. Training increased knowledge and confidence, and supported caregivers to reflect on how personally held values impacted practice. Identified training gaps were how to: (1) Discuss SHR with LAC demonstrating problematic sexual behaviours. (2) Record the SHR discussions that had occurred in LAC's health plans. Contrary to previous findings, caregivers regularly discussed SHR with LAC. Competing demands on time resulted in prioritisation of discussions for sexually active LAC and those 'at risk' of sexual exploitation/harm. Interagency working addressed gaps in SHR provision. SHR discussions placed emotional burdens on caregivers. Caregivers worried about allegations being made against them by LAC. Managerial/pastoral support and 'safe care' procedures minimised these harms.

**Conclusions** While acknowledging the existing level of SHR promotion for LAC there is scope to more firmly embed this into the role of caregivers. Care needs to be taken to avoid role ambiguity and tension when doing so. Providing SHR policies and training, promoting interagency

### Strengths and limitations of this study

► Looked-after children (LAC) are often excluded from school-based sexual health and relationships (SHR) education. The qualitative data generated from this study provide valuable insight into alternative means through which SHR discussions can be undertaken within this population.

► Participant accounts provided rich qualitative data that provided insight into how the introduction of SHR training and policies affected the undertaking of SHR discussions with LAC.

► Role theory underpinned the analysis and provided insight beyond existing research that is dominated by atheoretical accounts of the facilitators and barriers of SHR promotion in care settings.

► While our study focuses on the voices of residential care workers, foster carers and social workers, it does not capture the experiences of kinship carers or parents whose children may be looked after in their family home under the terms of a home supervision order.

working and providing pastoral support are important steps towards achieving this.

## INTRODUCTION

Children who are 'looked-after' are an important group for public health intervention. In the UK a child is considered to be a 'looked-after child' (LAC) if parental responsibility is assumed by social services or shared between the child's parents and social services. LAC can experience a variety of care arrangements, including: foster care; placement in residential (children's) homes, residential schools or secure units; being looked after by relatives or family friends in kinship placements; and being looked after at

home with their parents under the supervision of social services.[1][2] In 2016–2017, approximately 100 000 children were looked after by local authorities in the four UK nations.[3–6] While children may enter care as a result of their own or parental illness/disability, the majority become looked after as a result of neglect or abuse.[6][7]

Childhood maltreatment and precare experiences characterised by poverty, family breakdown, domestic violence, and parental mental illness and/or substance abuse increase LAC's vulnerability to experiencing: mental health difficulties; school exclusion; low educational attainment; substance use; and antisocial/offending behaviour.[8–11] Cumulative exposure to these factors, which are independently associated with increased sexual risk-taking,[12–16] mean LAC experience poorer sexual health outcomes than adolescents who have never been looked after. For example, studies from the UK, USA and Sweden have demonstrated increased rates of early sexual activity, greater numbers of lifetime sexual partners, poorer contraceptive use, and increased rates of sexually transmitted infections (STIs), teenage pregnancy/parenthood, sexual exploitation/assault and rape.[17–23]

The sexual health vulnerability of LAC may be exacerbated by limited access to sexual health and relationships (SHR) education. Barriers to SHR education include increased school absence/exclusion rates, frequent placement moves and poor-quality relationships with parents.[24–26] SHR discussions in care settings are also limited.[24][27–29] For example, a survey of Californian social workers reported that 66%–77% had never spoken to LAC about puberty, the characteristics of healthy relationships, abstinence, contraception, sexual orientation, STIs, pregnancy/parenthood and abortion.[28] Reported barriers to caregivers discussing SHR include: lack of knowledge and training; difficulties reconciling personal religious/moral values; lack of guidance about who is responsible for discussing SHR with LAC; uncertainty as to whether such discussions infringe on the rights of biological parents; concerns that discussing sex could be perceived as condoning sexual activity; uncertainty about the boundaries between confidentiality and child protection; and concerns about LAC making allegations of wrongdoing against caregivers.[27][28][30][31]

Calls have been made to improve LAC's access to SHR education.[32][33] Existing evidence shows that adolescents whose parents discuss SHR with them are more likely to delay intercourse, use contraception and have fewer sexual partners.[34–37] It has thus been recommended that caregivers discuss SHR with LAC. In Scotland, these recommendations have resulted in social work departments developing SHR policies and working with health boards to provide training to caregivers.[38][39] Key to this is changing the roles of those with legal responsibility for children in the care system.

To date, role theory[40–42] has been developed and applied mainly in healthcare settings. Research indicates that health practitioners develop specific beliefs and attitudes about their profession which establish the boundaries of their role and how they interact with other professionals, and those they serve.[43] Introducing new responsibilities can challenge these beliefs and blur the boundaries of existing roles.[44] While role blurring has advantages, such as allowing practitioners to adopt new responsibilities and be more responsive to the needs of the population they are working with, it can result in them feeling overwhelmed and uncertain about the limits of their new role.[45] Role blurring can also introduce sources of conflict, additional workload and ambiguity to the professional role.[43][46]

Role conflict occurs when practitioners attempt to fulfil two or more roles with incompatible expectations, requirements, beliefs or attitudes. For example, healthcare practitioners with strong personal views about the acceptability of abortion may find it difficult to provide information or advice about terminating an unwanted pregnancy.[47][48] Within the literature, role conflict is identified as a chronic source of stress that can result in healthcare professionals experiencing psychological distress.[49] Role overload occurs when the demands of a particular role exceed the capacity of the individual. For instance, through not having sufficient time to undertake tasks or through lacking the knowledge and skills necessary to undertake that role.[43][46] Finally, role ambiguity occurs when there is a lack of guidance/clarity about roles, and can result in: practitioners being unclear about what is expected of them; create concerns about duplication of work; and, act as a source of frustration, tension or anxiety at work.[46]

In this study we extend the application of role theory[40–42] to care settings, and more specifically, as a framework to answer the following research question: 'how do personal and institutional factors promote or limit the experience of role ambiguity, conflict and overload among caregivers tasked with discussing SHR with LAC?' In addressing this question we further develop the existing evidence base by moving beyond identifying barriers to SHR discussions being undertaken within the care system[24][27][29] and providing new insights into how SHR discussions can be incorporated into practice.

## METHODS

In-depth individual qualitative interviews were conducted with 22 caregivers from August to October 2011. As we wanted to explore how SHR discussions were influenced by caregivers' perceptions of their role, our sample included three groups of caregivers: foster carers (n=9), residential carers (n=8) and social workers (n=5).

### Recruitment and sample characteristics

All caregivers were recruited from a large urbanised local authority in Scotland, which had recently introduced SHR training for caregivers. In order to minimise service disruption, it was agreed with the local authority that sampling of social workers would be restricted to those based within one of the three geographically based

teams providing care to LAC. Sampling of foster carers was restricted to foster carers supervised by social workers working within that team. Residential carers were not geographically recruited; however, in order to minimise disruption it was agreed that unit managers, rather than front-line residential care staff, would be approached.

Respondents were randomly selected from staff lists stratified by whether caregivers had received SHR training or not. Our aim was to interview caregivers when half of all foster carers, residential carers and social workers had received training as it was felt that doing so would more accurately capture the role that training had on the inclusion of SHR discussions for each of the different caregiving roles. However, institutional delays in rolling out training meant that interviews were undertaken when half the foster carers, all residential carers and no social workers had received training. Our achieved sample reflects this.

The caregivers recruited were predominantly female (n=19); reflecting the gender imbalance seen within social services in the UK.[50] Participants' age was not formally recorded; however we know from ethnographic notes describing participants that foster carers were generally older (estimated age range of mid-40s to mid-60s) than residential carers and social workers (estimated age range early 30s to mid-50s). At the time of interview, the number of children living in placement was one to three for foster carers and four to eight for residential carers. Information about average case loads among social workers was not obtained, although we know from remarks made by one social worker during interview that she was managing 34 children aged 0–16 years. In 2009 a freedom of information request reported the average case load among social workers was seven.[51]

### Interview process
Interviews focused on: SHR training; experiences of providing advice about SHR; perceived barriers/facilitators to undertaking SHR discussions; which SHR topics were discussed; and how discussions were shaped by the experiences of LAC and their caregivers. A copy of the interview schedule can be found in online supplementary file 1.

Interviews were undertaken by the lead author (CN). Foster carers were interviewed within family homes without children and young people present, while residential carers and social workers were interviewed within private offices in residential children's homes and social work offices. Participants were not compensated for their time at the request of the local authority; however, they were afforded time within their working day to participate in interviews. On average, interviews lasted 45–60 min. Interviews were digitally recorded and transcribed verbatim.

### Data analysis and management
Prior to analysis, transcripts were compared with original recordings to ensure that narratives had been accurately captured, with any mistakes or missing segments updated by CN prior to coding being undertaken. Fieldwork notes on body language, facial expressions and emotional responses to questions were added to transcripts to supplement textual meaning. Transcripts were then anonymised and entered into NVivo V.9 (QSR International, Melbourne, Victoria, Australia) for data management and coding.

Thematic analysis was used to analyse the interview data generated. An initial coding framework was developed by CN based on both predetermined research questions and themes identified through close reading of all interview transcripts. The coauthors of this paper (MH and LE) reviewed the initial coding framework, along with a selection of transcripts, to identify any additional themes that they felt were missing from the initial coding frame. As part of this process, coauthors identified theories that they felt could be used to further refine the coding framework, with role theory[40–42] identified as a potential means of understanding how introducing SHR provision into the professional role impacted on caregivers' construction of their professional and personal identities.

The coding frame was revised by CN to reflect caregivers' experiences of role ambiguity, conflict and overload. Themes were also coded according to whether they acted as barriers or facilitators to the inclusion of SHR discussions within the professional role. At this stage, deletion and consolidation of themes was also undertaken. The finalised coding framework was agreed by all authors, before being applied to all manuscripts by CN. This allowed systematic comparisons to be made across the data and facilitated the identification of recurrent themes. During the coding process it was agreed by the authors that data saturation was occurring, with similar descriptions and narratives being presented by participants. Thus, further data collection was deemed unnecessary.

### Ethical approval
The research tools were reviewed and approved by the Director of Children and Family Services on behalf of the local authority. Standards for Reporting Qualitative Research (SRQR) reporting guidelines[52] were used to oversee the design, analysis and reporting of findings; see details in online supplementary file 2.

### Patient involvement
Patients were not involved in the design of this study.

### RESULTS
Our results highlight that inclusion of SHR discussions within caregiving is influenced by the interplay of institutional factors and caregivers' personal values/experiences. Table 1 provides a thematic overview of these factors and how they promoted or limited SHR promotion by contributing to the experience of role ambiguity, conflict and overload among caregivers. Illustrative

**Table 1** Qualitative mapping of how institutional and personal factors act as barriers/facilitators of sexual health and relationships (SHR) discussions and contribute to role ambiguity, conflict and overload among caregivers

| | Institutional or personal level factor | Barrier/facilitator of SHR discussions | Effect on caregiving role |
|---|---|---|---|
| **Role ambiguity** | | | |
| **Provision of sexual health policy and training reduces role ambiguity** | | | |
| A1. Challenges perceived taboos about discussing sex | Institutional | Facilitated SHR discussions by providing guidance and training | Reduced role ambiguity by emphasising that caregivers could and should discuss SHR with LAC |
| 'We have policies that we follow now in terms of sexual health, and it's something that's been brought to the forefront, where it's no considered taboo' (Shona, social worker/relief residential carer) | | | |
| 'In this industry, social services, there's been a lot of taboo about discussing sexual health. It's a priority now and it's seen as part and parcel of anything. I think that's cos of the policy and training' (Mary, social worker) | | | |
| 'It frees it up. You feel that, aye, it's no this big thing that shouldnae [shouldn't] be talked about. Children need to learn. They need to know that and we need to stop making it this big thing that they need to thing out themselves' (Pat, foster carer) | | | |
| A2. Emphasises corporate parenting responsibilities | Institutional | Facilitated SHR discussions by providing guidance and training | Reduced role ambiguity by clarifying expectations around caregiving role |
| 'We are corporate parents and we would do it with our own kids' (Joanne, residential carer). | | | |
| 'You would talk to your kids about it (sex). And that's what we do as corporate parents. We take on that role and responsibility' (Rachel, social worker) | | | |
| 'If we don't discuss it with a child, I think educate is too strict a term, but if we don't make them aware of it, then how are they gonna (going to) know?' (Shona, social worker/relief residential carer) | | | |
| A3. What it means to be a 'good' corporate parent | Personal | Facilitated SHR by promoting personal involvement in professional task | Reduced role ambiguity as a result of policy focus reflecting personal beliefs about parenting |
| 'We have a corporate parenting responsibility to all our kids but the key word there is parenting. Any good parent would spend time with their children talking about what is appropriate, when it is appropriate and how they should find out more information. As a parent you are trying to encourage young people to discuss with you that they've got partners, that they are engaged in sexual relationships, that they have got a girlfriend or boyfriend or whatever it may be... It's about reassuring young people that I am asking the exact same questions I ask my own daughter or my own son. You aren't being treated any differently because you live in a unit' (Mark, residential carer) | | | |
| A4. Policy reflects what usually happens within families | Personal | No perceived effect reported by caregivers | No effect |
| 'We don't shy away. If there was a sex scene in a movie or whatever, we quite often discuss it rather than say 'oh my goodness, we shouldn't be watching that, hide your eyes boys!' We just relax about it. It's not something that we try to pretend isn't there' (Alison, foster carer) | | | |
| 'The real conversations we have in here at tea times– and it always starts off with something silly. I had... was it a banana and custard yoghurt I had yesterday? And that started it off – one of them says: 'oh, that looks terrible' and this girl says 'but you can't determine whether you like it or not by looking at it. You've got to taste it'. 'No, I don't'. And this taste thing went on and on and it got into a discussion of peer pressure, didn't it? How it got there, I don't know. We just sit at the table through there and talk' (Ian, foster carer) | | | |
| A5. Clarifies expectations of role | Institutional | Facilitated SHR discussions by providing guidance and training | Reduced role ambiguity by clarifying expectations around caregiving role |

Continued

**Table 1** Continued

'Jane [policy developer] came to the unit manager's meeting and was kind of promoting young people's sexual health, what was our responsibility and where did we see our responsibilities being. And the training was very informative. It was very informative and made us look at our own sexual health and relationships. It gave us the tools to go away and... have these discussions wi' (with) young people' (Patricia, residential carer)

'We've never had any form of policy or training towards sexual health. It's something that as a manager I can say to them 'you've been given the information, you've been given the tools to put it into practice now' (Mark, residential carer)

"The training was mainly sharing stories about sexual health. You know, would you get a young person the morning after pill and what is age appropriate for that? I think we've progressed. A few years ago I was on holiday with a young person and I took her for the morning after pill and I had my bum booted... 'You shouldn't have done that. It wasn't your decision to take. Fortunately we've moved on as a department" (Joanne, residential carer)

**Lack of guidance contributes to role ambiguity**

| | Institutional or personal level factor | Barrier/facilitator of SHR discussions | Effect on caregiving role |
|---|---|---|---|
| A6. Lack of clear guidance on recording/reporting procedures | Institutional | Perceived barrier to LAC approaching caregivers for help and advice | Contributed to role ambiguity by creating confusion about how best to confidentially record SHR discussions |

Recording it is very difficult. We have general comms [communication] books which are for everybody's viewing, which is not appropriate, and we have individual logs which aren't appropriate either because the kid is maybe keen to keep something in confidence but then it is written down. It is a grey area' (Mark, residential carer)

'I had a LAC review where there young person, there was issues in terms of she was menstruating and leaving dirty sanitary towels and pants, like planting them places and hiding them. So I had written the report and I made a comment about some hygiene issues and said to my manager beforehand. There was a reason I had made it really vague as I didn't want to embarrass her. Unfortunately the foster carer decided to start talking about it and the girl burst into tears... and what I suppose I'm trying to highlight is that we need to be sensitive to the young person (Agnes, social worker)

**Role conflict**

***Balancing competing demands of child protection and preventative SHR work***

| B1. Monitoring sexual behaviour acts as a barrier to undertaking SHR discussions | Institutional | Barrier to preventative SHR discussions being undertaken due to focus on risk management | Contributed to role conflict |

'Safety is paramount' (Jane, residential carer)

'I was no longer a caregiver – I was a security guard. Keeping young ones out of other one's rooms that weren't supposed to be there, hauling other ones in windows that were trying to get out in the middle of the night, keeping ones out that didn't belong to the unit. We had fifteen year olds that we were hauling out of one room into another and saying 'No. You're not on'" (Karen, foster carer/former residential carer)

| B2. Undertaking SHR discussions in response to risk rather than preventatively | Institutional | Barrier to preventative SHR discussions being undertaken due to focus on risk management, and facilitated SHR discussions in response to risk-taking by LAC | Contributed to role conflict |

'There wouldn't always be a major, in-depth discussion if there weren't any major issues... but if a child is sexually active and they're underage, and... running away, having sex wi' men they don't know, coming back the next morning covered in mud, drinking... it would be very high on the agenda' (Agnes, social worker)

Continued

**Table 1** Continued

| | Institutional or personal level factor | Barrier/facilitator of SHR discussions | Effect on caregiving role |
|---|---|---|---|
| "He'd put on his profile something like 'I'm in care and I'm looking for...' One of the older girls had seen his profile and asked him right out in front of us 'why have you got that one your profile'. He was mortified. But that gave us the opportunity to sit down and tell him the reasons why he should have things like that on there. And even if you are gay, it's not the way you would word it, and it was actually our 16year old who said 'cos you don't know who is sitting looking at that profile and thinking oh he's game'" (Joanne, residential carer) | | | |
| B3. Strategies to manage the sexual health of LAC: monitoring relationships | Institutional | Facilitated SHR discussions about appropriate and positive relationships | Reduced role conflict |
| 'We have a young female (16) who is pregnant and her boyfriend (23) lives locally. He had been over for dinner and he has been involved in the unit and staff have met him and we are clear what our role is. It was quite clear to us that the best way for us to deal with it was to be part of the relationship. I was quite clear that in my role of safeguarding this young girl we had to get to know this young male and find out if there was any ulterior motive or if there was any reasons why he was interested in her, other than you know, a love for each other. So we engaged with him… We have been to his house on a couple of occasions, and we have met him with his mum as well. (Mark, residential carer) | | | |
| 'You are trying to encourage the young people to discuss with you that they've got partners and to bring them in so that we know them as a face round the unit. They're not allowed in bedrooms obviously, but they're allowed in the living area with the door open. And I would definitely encourage that unless I thought it was a negative influence' (Joanne, residential carer) | | | |
| B4. Strategies to manage the sexual health of LAC: monitoring phone and computer usage | Institutional | Facilitated SHR discussions about internet safety and sexual exploitation | Reduced role conflict |
| 'I'm no' that good at it, but we went into his facebook and realised the chats he's been having so we've started to speak about safety issues, you know, telling him that this person could be roond (round) the corner fae (from) you. It's a web cam' (Claire, residential carer) | | | |
| 'There was inappropriate material found on her phone, and in the past she's had images sent to her from people that in my opinion are grooming her, but she doesn't accept that she's at risk… So now we've got monitoring sheets. We monitor every shift what kids are doing on the computer and sometimes we think it's a wee bit of an overkill and obviously our internet is kind of sitting in the living room, very open, but we keep a very very very close eye… especially when you think they are at risk' (Joanne, residential carer) | | | |
| B5. Strategies to manage the sexual health of LAC: risk assessing outings | Institutional | Barrier to preventative SHR discussions being undertaken due to focus on risk management | Reduced role conflict |
| 'Last summer we stopped taking him to the play park… because he goes to younger children and he wants to pat them and cuddle them. I don't know if he is sexually aware… but he is almost compelled to do it… and he will sneak about to try and get to a wee one to give them a wee pat. So how do you deal with that? We stop taking him' (Pat, foster carer) | | | |
| 'If you've got child protection issues where you've got a young person who's maybe been sexually abused, and then sexually abused younger people, then we have to be dead strict as protecting other young people is also protecting them… I cannae (cannot) let him run aboot (about) just doon (down) the road because there's a wee nursery doon (down) the road. I cannae just let him go swimming. There's a whole protection risk assessment to which there' (Patricia, residential carer) | | | |
| B6. Strategies to manage the sexual health of LAC: managing space and room allocations | Institutional | Facilitated SHR discussions about privacy | Reduced role conflict |
| 'We had a serious incident where Craig (13) accused John (8) of more or less sexually abusing him. John was saying things like 'sex, sex, sex' and making thrusting movements because he knew it was upsetting Craig.… Craig couldn't deal with it. We found him urinating on John's bed and then he made this accusation. It was a terrible time for us all, only for it to turn out that Craig had made the whole thing up… as he wanted John moved. So we're very aware now of the two boys being separate. Craig sleeps upstairs and he has his own space up there. John is downstairs in a room along the corridor, and he is not allowed upstairs at all' (Alison, foster carer) | | | |

Continued

**Table 1** Continued

| | Institutional or personal level factor | Barrier/facilitator of SHR discussions | Effect on caregiving role |
|---|---|---|---|
| 'We've got one young person who most definitely has been sexually abused… and she can display quite predatory behaviour (later clarified by the caregiver stating that the young woman had been groomed into recruiting other LAC for a sex ring). She would encourage the rest of the group to go out drinking, and then make allegations of rape against one or more of the boys… We need to protect her and we need to protect others from her exposing them to inappropriate sexual contact for their age. That's something that we balance all the times in terms of the safety of the group. And that's how we decided her bedroom was best placed in close relation to the office' (Joanne, residential carer) | | | |
| ***Role conflict as a source of caregiver strain*** | | | |
| B7. Emotional impacts on caregivers | Personal | No perceived effect reported by caregivers | Consequence of role conflict |
| 'It's soul-destroying tae (to) try and stop that pattern of behaviour where young people would go met their pals… and be picked up by men that were pimping them… for a packet of cigarettes or a wee bag of sweeties. There would be times when they didnae (did not) want to have sex but they were forced and they would come in wi' pretty bad bruising and faces had been punched… It's pretty hard at times, but I think you've got tae be and be professional and say 'we're trying our best… sometimes we just don't succeed' (Patricia, residential carer) | | | |
| 'I had been away shopping and I came back in. The other boy was watching television and he seen me and goes 'I think you should go up the stairs'. Now this has happened on a few occasion, you know? If one of us has been out or distracted they would use that moment. I just put my bag down, didn't even take my coat off, and I ran up the stairs (Jean is visibly shaking and obviously upset). Here was child A and B in the sliding wardrobe, a pillow put down on the inside of the sliding wardrobe. He had the girl on the floor on the top of her… that's how quickly' (Jean, foster carer) | | | |
| **Concerns about the potential for false allegations being made by LAC** | | | |
| C1. Discussing SHR places caregivers in a position of vulnerability | Personal | Barrier to SHR discussions arising from caregivers' concerns about their own vulnerability | Contributed to role conflict |
| 'I had to leave the room and when I came back my manager was like 'I needed to come out' and basically he'd been sitting and the boy (who had had been groomed and sexually abused by a paedophile ring) had got an erection. He felt really uncomfortable cos obviously he was on his own with him and he didn't want to be on his own with him… so he got up and walked out. As workers we can be quite vulnerable… so we have to be very aware of how we protect ourselves (Agnes, social worker) | | | |
| 'You imagine right that one of the young persons' approached you right and said 'I'm thinking o' having sex wi' my boyfriend. What do you think? And then the next night the nightshift comes on and you're away and they say 'guess what she was saying tae me last night. Aw she was doing was talking about sex'. That can be misconstrued and before you know it it's a big fact finding investigation' (Patricia, residential carer) | | | |
| It's worrying… my son's a police officer, my husband works in law enforcement and I work with students – so given that we all have to be vetted and disclosed at work - we have to take extra care.' (Alison, foster carer) | | | |
| C2. Strategies to protect caregivers against false allegations: recording conversations | Institutional | Facilitated SHR discussions by providing a safer environment for caregivers | Reduced role conflict |
| 'Because of the risk that it presents to them as workers in terms of possible allegations or comments being made in future… we need to make sure that any information we are sharing with young people is appropriately recorded, accurately recorded… And if there is anything inappropriate, you know I am thinking, you know, maybe a female resident making a comment to a male member of staff then that's been appropriately recorded and raised and that the staff member and the young person are both supported and discussions are held about what is appropriate and what is not' (Mark, residential carer) | | | |
| 'I think that one of the things we had to obviously highlight was Safe Care and the recording of that sensitive conversation… how do you have that conversation in an environment where you're safe? Because if you're talking about closed doors she could make an allegation against you. So it's about recording the discussion you had. You don't have tae dae War and Peace but 'she came and she asked me aboot this and this was the advice I gave her" (Patricia, residential carer) | | | |

Continued

**Table 1** Continued

| | Institutional or personal level factor | Barrier/facilitator of SHR discussions | Effect on caregiving role |
|---|---|---|---|
| C3. Strategies to protect caregivers against false allegations: having someone else present | Institutional | Facilitated SHR discussions by providing a safer environment for caregivers | Reduced role conflict |

'John had a wee urine infection and his penis was so sore, so it was a case of 'well, let's have a wee look and see if it's all red'… he's comfortable with that and it's all fine, but as a foster carer I'm not gonna go into a room and close a door and have a look at a 10year old's penis. I'm gonna say 'right, Mark (husband) and I will sit on the bed and you touch it. You show me' and then 'right, ok, here's some cream' (Alison, foster carer)

'You would wait until the house was quieter and maybe do some of that work. 'Why don't we go on the computer next door and we'll shut the dining room door over' but I'll have a member of staff going in and out of the kitchen' (Anna, residential carer)

| | Institutional or personal level factor | Barrier/facilitator of SHR discussions | Effect on caregiving role |
|---|---|---|---|
| C4. Strategies to protect caregivers against false allegations: household rules | Institutional | Facilitated SHR discussions about privacy and boundaries | Reduced role conflict |

'You need to keep reinforcing what is and is not appropriate behaviour… it is not appropriate to be showing yourself off. It's not appropriate to be going into the toilet with other boys' (Karen, foster carer)

'The wee things that you don't actually think about change, because, you know, it was quite natural for our boys to come down in the morning in their boxer shorts – maybe wae a dressing gown in the summer, maybe not. That changes. That stops. All that stops. You, you have to look at all the risks there are, and your, your children's life changes. Our 7year-old couldn't come and jump into our bed in the morning because I couldn't allow the other two children to do it – so I couldn't allow him to do it because I didn't want them to feel that he was special' (Pat, foster carer)

**Personal values and experiences**

| | Institutional or personal level factor | Barrier/facilitator of SHR discussions | Effect on caregiving role |
|---|---|---|---|
| D1. Religious and moral values as a source of role conflict | Personal | Barrier to SHR discussions, particularly those focused on sex, sexuality and abortion | Contributed to role conflict |

'I'm a practising Catholic. I don't hold the church in any great high esteem but I have faith and as a parent myself I have never brought my children up tae… all this input of you can go get the pill here, you can go get a jag here and here's what all that's about in such graphic detail. I know this sounds as if its' so traditional and old fashioned but I was never brought up with all this input. I suppose I'm still traditional in my own family home' (Claire, residential carer)

'I've got a Catholic upbringing and you didn't do anything until you were married. It was very strict. I wouldn't force that (talking about sex) on any of the kids that I work with' (Anne-Marie, foster carer)

'Faith based values among staff can sometimes act as a barrier to workers discussing sexual health with young people' (Joanne, residential care)

'It has took me an awful long time tae do all my challenging in myself and asking and prying about [about] how does that fit with my psyche to sit here and talk about things that I ordinarily would not talk aboot. I went on that course and I found it so challenging. 'Why are we no' talking about sex to these weans'? Why are we no' talking about relationships?' And I got in this pure big debate wi' myself: 'I wouldnae tell my boy that. I wouldnae tell my lassies that' and the trainer was really helpful with me and saying 'yeah, but you need to remember that these kids arenae getting' what your own kids are getting' (Claire, residential carer)

'There can be a clash between what workers may want and what the city council wants…so the training was very much to do with looking at our values, our value base and our knowledge… what was very surprising was the fact that kids are learning so much younger, we were like 'oh my goodness, kids are talking about that (sex) at such a young age' (Laura, residential carer)

| | Institutional or personal level factor | Barrier/facilitator of SHR discussions | Effect on caregiving role |
|---|---|---|---|
| D2. Being allowed to challenge and reflect on values in training | Institutional | Facilitated SHR discussions by challenging pre-existing beliefs and emphasising vulnerability of LAC | Reduced role conflict |

Continued

**Table 1** Continued

| | Institutional or personal level factor | Barrier/facilitator of SHR discussions | Effect on caregiving role |
|---|---|---|---|
| *'It has took me an awful long time tae do all my challenging in myself and asking and prying aboot [about] how does that fit with my psyche to sit here and talk about things that I ordinarily would not talk aboot. I went on that course and I found it so challenging. 'Why are we talking about sex to these weans? Why are we no' talking about relationships?' And I got in this pure big debate wi' myself: 'I wouldnae tell my boy that.' I wouldnae tell my lassies that' and the trainer was really helpful with me and saying 'yeah, but you need to remember that these kids arenae getting' what your own kids are getting'' (Claire, residential carer)* | | | |
| *'There can be a clash between what workers may want and what the city council wants...so the training was very much to do with looking at our values, our value base and our knowledge... what was very surprising was the fact that kids are learning so much younger, we were like 'oh my goodness, kids are talking about that [sex] at such a young age''* (Laura, residential carer) | | | |
| D3. Pastoral support as a means of supporting caregivers experiencing role conflict | Institutional | Facilitated SHR discussions by providing caregivers with support to discuss challenging topics or through providing LAC with access to another caregiver to discuss issues with | Reduced role conflict |
| *'One of the foster carers I work with, she's never been used to talking to children about sex in any way and she asked me to undertake that as I had went on the training'* (Anne-Marie, foster carer) | | | |
| *'She was really struggling with the fact that one of the girls in her care had approached her and told her that she was pregnant, but wanted to terminate the pregnancy. She was Catholic and very uncomfortable. My view was that this worker already had a relationship with this girl so it was my job to support her to present all the options to her'.* (Joanne, residential care) | | | |
| D4. Own experiences of sexual health and relationships as a motivator for discussing SHR | Personal | Facilitated SHR discussions by motivating caregivers to ensure that LAC received better access to information than they had during childhood | Reduced role conflict |
| *'I went to college at sixteen and... I'm sitting in a class and I'm looking at this film on childbirth and I see where a baby's born from. I thought that they untied your tummy button, took it out, tied it up again and stuck it back in. Now I did bring my children up... from when they were wee tots... I would get them to go and get my sanitary towels and I would tell them what it was'.* (Pat, foster carer) | | | |
| *'We got told nothing, absolutely nothing, to the stage where the first time I took a period I thought I was dying. And then when I had my first baby I didn't have a clue what was happening or what was going to happen to me so I always thought that if I had children of my own I would prepare them* (Anne-Marie, foster carer) | | | |
| D5. Having 'parented' around sex | Personal | Facilitated SHR discussions by providing caregivers with parenting experiences to draw on | Reduced role conflict |
| *'It was always just a natural kind of growing up. We spoke about contraception, and my daughter, I was able to go with her to the doctors when she wanted to start taking the pill. We could just talk about it really openly. Likewise, with John (foster child), we've approached the subject of puberty and changes in the body'* (Alison, foster carer) | | | |
| **Role overload** | | | |
| **Workforce capacity** | | | |
| E1. Limited staff numbers in residential care | Institutional | Barrier to SHR discussions due to focus on risk management and having to prioritise resources | Contributed to role overload |

Continued

**Table 1** Continued

| | Institutional or personal level factor | Barrier/facilitator of SHR discussions | Effect on caregiving role |
|---|---|---|---|
| 'We're limited wi' [with] staff. We should have two on every shift so if you had a member of the team who was doing that work maybe 2–3 hours a week there is an impact on the other five young people you're looking after' (Patricia, residential carer) | | | |
| 'Children's unit staff are really well-placed to do stuff like that [discuss SHR]. They should be able to spend the time, but sometimes it doesn't seem to happen. I don't know why. I don't know if they're caught up in paperwork and ordering things, and dealing with incidents that have happened' (Louise, social worker) | | | |
| 'I worked in a 19–23 bedded unit and it was, the work was mostly chaotic. It was like firefighting and you were just going in and trying to contain your shift' (Joanne, residential carer) | | | |
| E2. Competing demands on social workers' time | Institutional | Barrier to SHR discussions due to caregivers having limited time to form trusting relationships with LAC | Contributed to role overload |
| 'I don't think we have the time to give young people the time they need and the support they need. That's just the way things are going to be. The service is just getting narrower… Sometimes you don't even have time to go to training as you get called to court' (Agnes, social worker) | | | |
| 'As a social worker it's a lot more difficult to really get to know that young person because in residential… you really get to know the young people because you see them for 24 hours periods, and you know a lot more about their life, and what's happening on a daily basis… being a social worker… there's a lot more hidden. You maybe find out a month later that something happened… and it's a lot more difficult to establish what. Spending time wi' young people and building up that relationship is what opens more doors to the speaking to you directly about it (SHR)' (Shona, social worker/relief residential carer) | | | |
| 'a safety plan gets planned and implemented… and focused work carried out that is specific and tailored to that young person's needs and risks… that's something that as the allocated worker I would review and monitor' (Mary, social worker) | | | |
| E3. The importance of interagency working to ensure that LAC receive SHR | Institutional | Facilitated SHR discussions by providing additional supports to undertake concentrated SHR work | Reduced role overload |
| 'We've got a 12year old girl… and all her talk and her chat is about paedophiles, and she was going on websites and there was inappropriate material found on phones so obviously our alarm bells are ringing… She's so vulnerable. We've still not got feedback from the police what was on the phone. We give her her wee trust exercises back on the computer but then she just tries to go onto these certain websites. She's had images sent to her from people that in my opinion are grooming her and she doesn't see that, she doesn't accept that she's at risk… So, we spoke to her worker at the young woman's project and she's covering a lot of that groundwork with her. And someone here is doing the work about keeping herself safe and making safer choices on the computer' (Joanne, residential carer) | | | |
| 'It's a bit aboot (about) sharing you know? We kind of all come together. Agency X does the risk assessment work, and they work wi' the young person about why it happened, their feelings, whatever. Agency Y work wi' him to provide socialisation – taking him out because obviously he's not allowed out unsupervised' (Patricia, residential carer) | | | |
| E4. Avoiding duplication of workload | Institutional | No perceived effect reported by caregivers | Reduced role overload |
| 'We've had reports from them and we know what they are doing… so we tend to back off and let one person do that work on sexual health and keeping safe' (Joanne, residential carer) | | | |
| 'Sharing of information is key' (Mary, social worker) | | | |
| **Workforce composition** | | | |
| E5. Low proportion of men working in residential care excludes male LAC from SHR discussions | Institutional | Barrier to male LAC accessing SHR information | Contributed to role overload for female caregivers |
| 'There's no gender balance in residential… for every hundred applicants I can guarantee you that about 84% of them are women' (Patricia, residential carer) | | | |

Continued

**Table 1** Continued

| | Institutional or personal level factor | Barrier/facilitator of SHR discussions | Effect on caregiving role |
|---|---|---|---|
| *'If there wasnae a male on shift then the boys wouldn't come and talk to us about sex'* (Laura, residential carer) | | | |
| *'I dinnae really think boys have really come and speak to ye as much as girls, but then again, they might be more likely tae speak tae like a male, like a male worker'* (Shona, social worker/relief residential carer) | | | |
| E6. Male caregivers better placed to talk to male LAC about sex-specific practices | Personal | Facilitated male LAC accessing SHR information when male caregivers were available to discuss issues | Reduced role overload for female caregivers |
| *'Teaching them how to shave for example, that's not something I can do. So, if I have a male worker, then I get him to come into work unshaven so he can show the boys how to shave properly'* (Trisha, residential carer) | | | |
| *'He was always pulling at himself, wasn't he? And I said, 'do you know something? You need, when you're in the shower, you need to get your penis, pull your foreskin back and clean it with soap and water'. And he just stood there, but it cured it, didn't it?'* (Ian, foster carer) | | | |
| **Not having sufficient skills and knowledge** | | | |
| F1. Caregivers identifying that they need specialist training to undertake SHR discussions | Personal | Barrier to discussing SHR with LAC due to caregivers' perceived lack of knowledge about SHR topics and how to discuss these with LAC | Contributed to role overload |
| *'if I was in the position of working with a young person who had a very trusting relationship with me, and who required support with their sexual health and development, then I would like to play a part in that… but I'd like training because I see that as a gap'* (Mike, social worker) | | | |
| *'For workers, especially for workers who are not used to working with teenagers there is a need for more formal training, and formal training more often… I mean we do refresher courses for other training, but I can't remember the last time I saw a sexual health awareness or sexual health programme* (Agnes, social worker) | | | |
| F2. Sexual health and relationships training as a source of knowledge | Personal | Facilitated SHR discussions by providing caregivers with SHR knowledge and the skills needed to discuss these with LAC | Reduced role overload |
| *'A lot of the training was about words you've not heard since you were a kid… we need to know what these kids mean when they are saying certain things'* (Joanne, residential carer) | | | |
| *'the course opens your eyes to it, you know? You can go through life thinking, well, right, ok, I know about Gonorrhoea and this kind of stuff, but I don't know about Chlamydia, and I don't know about this, that and the next thing. And these are all things that children can get, and I need to be able to explain what can happen if they have unprotected sexual relationships'* (Ian, foster carer) | | | |
| *'one worker talked about your flower, and if you needed anything sorted you would go to the flower shop… I don't think that things like that really help when talking about going to clinics and your vulva… You need to use the proper names so that everyone is quite clear she could have people thinking 'oh right, I need to go buy some flowers'… because they take you literally'* (Laura, residential care) | | | |
| F3. Sexual health and relationships training as a source of confidence | Personal | Facilitated SHR discussions by promoting confidence and reducing embarrassment among caregivers | Reduced role overload |
| *'After I went to the training I found that I was really more confident and I had all the information on hand and booklets to show to the boy… and he said to me at the very end that he'd been having sex education at school, but that I had explained it far better. I put that down to the training'* (Anne-Marie, foster carer) | | | |

Continued

**Table 1** Continued

| | Institutional or personal level factor | Barrier/facilitator of SHR discussions | Effect on caregiving role |
|---|---|---|---|
| *'the training has definitely equipped the staff with confidence'* (Tricia, residential carer) | | | |
| F4. Using sexual health promotion materials | Institutional | Facilitated SHR discussions by providing caregivers with resources they could access and use with LAC | Reduced role overload |
| *'If I've no got an answer for them I'll maybe say 'we've got literature on that so just gie me a minute and we'll go and get it and we'll take 5 min to go through it'* (Patricia, residential carer) | | | |
| *'to be that if you don't know it, don't pretend that you do but let the kids know that, 'well, look, I don't know about that, but I've got a phone number I can phone'* (Ian, foster carer) | | | |
| F5. Training highlighted that SHR discussions were routinely happening in care | Personal | No perceived effect reported by caregivers | Contributed to role overload in some cases by creating anxiety that SHR information being provided was correct |
| *'we were playing Connect 4, and one of the girls said 'how do you get pregnant' and we said 'well, you need to have sex'. 'Aye, I know that… and I know that he cums, but how does that then work?' So we dismantled the Connect 4, and we said 'well it's no square, but you'll have to imagine this is a womb, and these are the fallopian tubes', and we used the wee circles as the sperms and the eggs, and we used that to explain it…. Once we were finished I turned to (another caregiver) and said 'did I get that right?''* (Claire, residential carer) | | | |
| *'I remember, certainly, a few years ago having a discussion with an 18 year old girl who wasn't sure what she looked like, ermm, err, down below. What her vulva looked like. And about sex, sexual intercourse. She didn't know whether she would be able to partake in that and I do, really remember, just frankly saying to her, why don't you just get a mirror and have a look, you know, oh I can't be doing that, but why can't you be doing that, it's the easiest way to kind of look and have a see, to explore your own bodies and you'll know what's likeable, what's not likeable, what you're happy with people to touch and what you're not happy for people to touch'* (Tricia, residential care) | | | |
| **Pastoral support** | | | |
| G1. Support of management | Institutional | Facilitated SHR discussions by providing caregivers with additional supports | Reduced role overload |
| *'I found that work really difficult, because I had never had to deal with… trying to manage a child – cos he was a child at the time – who is not only, you know, being abused, but is an abuser… I felt really, you know, unsure of how best to manage that. One of the best things with managing that was that my manager agreed to support me, and we did the work together'* (Agnes, social worker) | | | |
| G2. Peer supervision | Institutional | Facilitated SHR discussions by providing caregivers with additional supports and continued informal learning | Reduced role overload |
| *'We quite often in this team have group supervision… where I might not have had the experience of working with a young person in that situation for a couple of months, someone else probably has or will have without doubt, so it's about other people sharing their experiences and information and sometimes that's the best way to learn because you are speaking about real experiences and examples (Agnes, social worker)* | | | |
| *'We deal with it pretty well, but I think with this wee core group of carers that we've got there's always an opportunity for learning… 'I've tried to get this boy to do his bloody homework and he just will not do it' and somebody will say 'try this' and you find that it works. That's where our support is… from other carers in our group. We bounce off each other' (Ian, foster carer)* | | | |

LAC, looked-after children; SHR, sexual and health relationships.

quotes, referred to by number (eg, A1), are presented within both the main body of results and table 1 as supportive evidence of the themes discussed.

## Provision of sexual health policy and training reduces role ambiguity

Role ambiguity occurs when there is a lack of guidance and clarity about roles, and can result in practitioners being unclear about what is expected of them.[46] In this study we found that the provision of SHR policies reduced role ambiguity and increased the acceptability of SHR discussions among residential carers and social workers. This occurred by creating an institutional-level ethos that challenged historical taboos about discussing sex and emphasised the inclusion of SHR discussions within caregivers' professional role. For instance: '*We have policies that we follow now in terms of sexual health and it's something that's been brought to the forefront, where it's no considered taboo*' (A1 - Shona, social worker/relief residential carer).

In particular, SHR policies emphasised that undertaking SHR discussions with LAC allowed caregivers to fulfil their 'corporate parenting' responsibilities: '*you would talk to your kids about it (sex). And that's what we do as corporate parents. We take on that role and responsibility*' (A2 - Rachel, social worker). Among residential carers and social workers, incorporating institutional expectations about corporate parenting responsibilities into practice resulted in their developing personally held views about what it meant to be a 'good' corporate parent. These views focused on how as caregivers they should ensure that LAC were provided with the same access to SHR information as children not in the care system. Hence, SHR discussions were identified as being part of routine caregiving (A3).

The importance of corporate parenting was not reported by foster carers, who instead discussed their personal views that SHR discussions were part of normal family life: '*we don't shy away. If there was a sex scene in a movie or whatever, we quite often discuss it rather than say 'oh my goodness, we shouldn't be watching that, hide your eyes boys!' We just relax about it. It's not something that we try to pretend isn't there*' (A4 - Alison, foster carer).

SHR policies and training clarified at an institutional level what was expected of caregivers when discussing SHR with LAC. For instance, one caregiver described how the individual within social work services with responsibility for implementing SHR policies and training had come along to a staff meeting to discuss the role that residential carers were expected to play in implementing it: '*Jane [policy developer] came to the unit manager's meeting and was kind of promoting young people's sexual health, what was our responsibility and where did we see our responsibilities being. And the training was very informative. It was very informative and made us look at our own sexual health and relationships. It gave us the tools to go away and… have these discussions wi' young people*' (A5 - Patricia, residential carer). SHR training was also identified as clarifying the range of topics that caregivers would be expected to discuss with LAC. This

clarification was welcomed by several residential carers who had been reprimanded in the past for doing things such as accompanying LAC to obtain emergency contraception (eg, A5 – Joanne, residential carer).

## Lack of guidance contributes to role ambiguity

While SHR policies reduced role ambiguity through clarifying what was expected of caregivers when discussing SHR with LAC, lack of institutional guidance on specific practice elements created confusion and contributed to role ambiguity. One example of this was how SHR discussions should be recorded, with residential carers perceiving that they did not have clear guidance about how to confidentially record and present information when there were no child protection concerns. For instance, one residential carer stated: '*recording it is very difficult. We have general comms [communication] books which are for everybody's viewing which is not appropriate, and we have individual logs which aren't appropriate either because the kid is maybe keen to keep something in confidence but then it is written down. It is a grey area*' (A6 - Mark). Uncertainty about how to record information in a way that protected the confidentiality of LAC was perceived as a potential barrier to LAC approaching caregivers for help.

## Role conflict

Role conflict occurs when practitioners attempt to fulfil two or more roles with incompatible expectations, requirements, beliefs or attitudes.[47 48] In this study we identified three examples of how including SHR interactions into caregiving practices resulted in three sources of role conflict that acted as barriers to SHR. These were: (1) The challenges of balancing the competing responsibilities of child protection concerns with undertaking preventative SHR discussions. (2) Caregivers' concerns about the potential for allegations being made against them by LAC influencing decisions about how and when SHR discussions were undertaken. (3) Challenges that caregivers faced in reconciling professional responsibilities for discussing SHR with personally held religious/moral views about sexual behaviours. These challenges, and the approaches that caregivers used to address them, will now be discussed.

### Balancing competing demands of child protection and preventative SHR work

Caregivers experienced role conflict as a result of trying to balance the competing institutional responsibilities of child protection concerns with undertaking preventative SHR discussions. Many of those interviewed stated that their role often focused more on policing/monitoring sexual behaviour than providing SHR information: '*I was no longer a caregiver – I was a security guard*' (B1 - Karen, foster carer/former residential carer). As the main institutional priority was to safeguard LAC, focussing on risk-taking meant that SHR discussions were often undertaken solely in response to specific safety concerns: '*there wouldn't always be a major, in-depth discussion if there weren't any major*

*issues… but if a child is sexually active, and they're underage, and… running away, having sex wi' [with] men they don't know, coming back the next morning covered in mud, drinking… it would be very high on the agenda' (B2 - Agnes, social worker).* This resulted in some LAC being excluded from SHR discussions.

The challenges of managing child protection risks were particularly emphasised by those caring for LAC who had experienced sexual exploitation/abuse and/or were demonstrating sexually inappropriate or sexually abusive behaviours towards others. Caregivers identified a range of strategies advocated by their institution to manage the sexual health of LAC, including: monitoring peer and romantic relationships (B3); monitoring mobile phone and computer usage (B4); assessing the risks associated with young people in their care going on both supervised and unsupervised outings (B5); and carefully managing internal public spaces and room allocations (B6). For example, one residential carer described how the space within residential care settings was carefully managed to provide additional monitoring and to protect residents: *'We've got one young person who… can display quite predatory behaviour [later clarified by the caregiver stating that the young woman had been groomed into recruiting other LAC for a sex ring]. She would encourage the rest of the group to go out drinking, and then make allegations of rape against one or more of the boys… We need to protect her and we need to protect others from her exposing them to inappropriate sexual contact for their age. That's something that we balance all the times in terms of the safety of the group. And that's how we decided her bedroom was best placed in close relation to the office' (B6 - Joanne).*

Having strategies to draw on appeared to reduce the experience of role conflict for caregivers by situating some SHR discussions within the context of risk management (eg, B4 - Claire).

### Role conflict as a source of caregiver strain

Although institutional supports were available for managing the sexual health of LAC, the constant monitoring of behaviour understandably affected caregivers' emotional well-being. For example, one foster carer became upset and started visibly shaking while recounting an incident where she had found two siblings who had been left unsupervised *'in the sliding wardrobe, a pillow put down on the inside to mask the noise… and the girl on the floor as he was on top of her holding her down' (B7 - Jean).* The perceived impacts of caregivers' emotional distress on SHR provision was not directly discussed.

### Concerns about the potential for false allegations being made by LAC

Being asked to discuss SHR with LAC was perceived as a task that placed caregivers in a position of vulnerability. For instance, one of the social workers described a situation where being left alone with a young man who had been groomed and sexually abused by a paedophile ring had left her line manager feeling very vulnerable: *'I had to leave the room and when I came back my manager was like 'I*

*needed to come out' and basically he'd been sitting and the boy had got an erection. He felt really uncomfortable cos obviously he was on his own with him and he didn't want to be on his own with him… so he got up and walked out. As workers we can be quite vulnerable' (C1 - Agnes).*

Feeling vulnerable resulted in caregivers expressing concerns that undertaking SHR discussions could result in false allegations of abuse being made against them by LAC. These concerns contributed to the experience of role conflict by highlighting to caregivers the potentially negative reputational and financial consequences that could occur if steps weren't taken to reduce the risk of allegations occurring: *'It's worrying… my son's a police officer, my husband works in law enforcement and I work with students – so given that we all have to be vetted and disclosed at work - we have to take extra care' (C1 - Alison, foster carer).*

Wanting to protect against false allegations affected how SHR discussions were undertaken, with all of those interviewed emphasising how following institutional guidelines and developing 'safe care' practices helped minimise the risk of allegations being made against caregivers. The practices adopted by caregivers included ensuring that discussions were accurately documented (C2), having conversations in private spaces with another caregiver present (C3) and developing household rules about things like personal contact, privacy and modesty (C4). These practices were also used to safeguard children and other family members living within the household (eg, C4 - Karen).

### Personal values and experiences

The institutional expectation that SHR should be discussed with LAC contributed to the experience of role conflict among caregivers who held religious and moral views about the appropriateness of openly discussing sexual behaviours. For instance, one of the foster carers described how her faith had impacted on views about her practice as a caregiver: *'I've got a Catholic upbringing and you didn't do anything until you were married. It was very strict. I wouldn't force that [talking about sex] on any of the kids that I work with' (D1 - Anne-Marie).*

Being provided with time in SHR training sessions to reflect on how personal values could affect SHR discussions was identified as one way that the institution could support caregivers in addressing this conflict: *'It has took me an awful long time tae do all my challenging in myself and asking and prying aboot (about) how does that fit with my psyche to sit here and talk about things that I ordinarily would not talk aboot' (D2 - Claire, residential carer).* Pastoral support was also identified as an institutional means of supporting caregivers experiencing difficulties reconciling personal views with professional expectations. For example, one residential carer described supporting a fellow caregiver who strongly opposed abortion to *'present all the options'* to a girl in her care who wanted to terminate the pregnancy (D3 - Joanne).

Although religious and moral views acted as a source of conflict for some caregivers, other experiences acted to

reduce role conflict by providing caregivers with personal motivation for incorporating SHR discussions into their professional practice. For example, caregivers' own experiences of not receiving school-based or home-based SHR education as a child reduced conflict by providing motivation for SHR discussions being incorporated into practice: '*I went to college at sixteen and… I'm sitting in a class and I'm looking at this film on childbirth and I see where a baby's born from. I thought that they untied your tummy button, took it out, tied it up again and stuck it back in. Now I did bring my children up… from when they were wee tots… I would get them to go and get my sanitary towels and I would tell them what it was*' (D4 - Pat, foster carer). Having discussed SHR with a biological child also helped to reduce conflict by providing caregivers with experiences that they could draw on when talking to LAC: '*it was always just a natural kind of growing up. We spoke about contraception, and my daughter, I was able to go with her to the doctors when she wanted to start taking the pill. We could just talk about it really openly. Likewise, with Michael [foster child], we've approached the subject of puberty and changes in the body*' (D5 - Alison, foster carer).

### Role overload

Role overload occurs when the demands of a particular role exceed the capacity of the individual. For instance, through not having sufficient time to undertake tasks or through lacking the knowledge and skills necessary to undertake that role.[43 46] Lack of capacity acted as a barrier to SHR discussions being undertaken by caregivers and was associated with the experience of role overload. Four instances of this were identified: (1) Workforce capacity, that is, not having sufficient time to discuss SHR with LAC. (2) Workforce composition. (3) Not having the necessary skills/knowledge to talk about sexual health. (4) Needing managerial and pastoral support to undertake SHR discussions. These barriers, along with the strategies used to address these, will now be discussed.

### Workforce capacity

Having insufficient numbers of caregivers acted as both a source of role overload and an institutional barrier to residential carers undertaking SHR discussions. For instance, one residential carer described the challenges that limited staffing presented in caring for a young man in placement who had both '*been sexually abused*' and '*sexually abused younger children*'. These challenges included balancing the needs of other children in placement with being able to provide him with appropriate opportunities for socialisation, monitor his behaviours and discuss the effects of previous abuse on his understandings of relationships: '*We're limited wi' [with] staff. We should have two on every shift so if you had a member of the team who was doing that work maybe 2–3 hours a week there is an impact on the other five young people you're looking after*' (E1 - Patricia). The number of children living in placement was identified as contributing to role overload, with larger group home settings presenting more challenges to staff (eg, E1 - Joanne).

Social workers also reported having limited capacity to discuss SHR with LAC. The main reason cited for this was that competing demands meant social workers did not have time to build up the trusting relationships LAC formed with residential carers, foster carers and key workers provided by external agencies: '*as a social worker it's a lot more difficult to really get to know that young person because in residential… you really get to know the young people because you see them for 24 hours periods, and you know a lot more about their life, and what's happening on a daily basis… being a social worker… there's a lot more hidden. You maybe find out a month later that something happened… and it's a lot more difficult to establish what. Spending time wi' young people and building up that relationship is what opens more doors to them speaking to you directly about it [SHR]*' (E2 - Lindsay, social worker). Institutional demands on social workers' time, along with the perception that families distrusted social workers, led to it being perceived that the role of the social worker should focus on coordinating and monitoring the promotion of SHR to LAC: '*a safety plan gets planned and implemented… and focused work carried out that is specific and tailored to that young person's needs and risks… that's something that as the allocated worker I would review and monitor*' (E2 - Mary, social worker).

To ensure that LAC were not excluded from SHR discussions due to limited capacity among caregivers to undertake these, all of those interviewed stressed the importance of interagency and partnership working. For instance, Patricia (residential carer) returned to her example of the young man who had been '*sexually abused*' and '*sexually abused younger children*' to describe how several agencies worked to support caregivers to monitor his behaviour and undertake SHR discussions with him: '*It's a bit aboot [about] sharing you know? We kind of all come together. Agency X does the risk assessment work, and they work wi' the young person about why it happened, their feelings, whatever. Agency Y work wi' him to provide socialisation – taking him out because obviously he's not allowed out unsupervised*' (E3). This was particularly valued when the SHR topics that needed to be discussed with LAC related to more specialised topics such as sexual exploitation and the adoption of problematic sexual behaviours which caregivers did not have sufficient time or training to focus on (eg, E3 - Joanne).

Partnership working had the potential to contribute to role overload by duplicating the workloads of caregivers. To avoid this, and to ensure LAC did not receive contradictory messages, caregivers highlighted the importance of being informed about the content of SHR education that was delivered by partner organisations: '*we've had reports from them and we know what they are doing… so we tend to back off and let one person do that work on sexual health and keeping safe*' (E4 - Joanne, residential carer).

### Workforce composition

The low proportion of male residential carers was identified as a potential barrier to male LAC receiving SHR information: '*If there wasnae [wasn't] a male on shift then the*

*boys wouldn't come and talk to us about sex' (E5 - Laura, residential carer).* There were two reasons for this. First, it was perceived by caregivers that LAC were more comfortable and less embarrassed discussing SHR with caregivers of the same sex (E6). Second, female caregivers perceived that male caregivers, as a result of their lived experiences, were often better placed to explain or demonstrate sex-specific hygiene practices: *'teaching them how to shave for example, that's not something I can do. So, if I have a male worker, then I get him to come into work unshaven so he can show the boys how to shave properly' (E7 - Trisha, residential carer).* Not having sufficient numbers of men working in residential care was identified as a potential source of role overload for female residential carers as a result of their having to take on the role of being predominantly responsible for undertaking SHR discussions.

### Not having sufficient skills and knowledge

Not having the skills and training needed to undertake SHR discussions contributed to the experience of role overload and acted as a barrier to such discussions being undertaken by caregivers. Being provided with SHR training by social work services was identified as a way to address this: *'if I was in the position of working with a young person who had a very trusting relationship with me, and who required support with their sexual health and development, then I would like to play a part in that… but I'd like training because I see that as a gap' (F1 - Mike, social worker).*

SHR training was reported to improve caregivers' knowledge of sexual health terminology, contraceptive choices, STIs and colloquialisms used by LAC to describe sexual acts: *'the course opens your eyes to it, you know? You can go through life thinking, well, right, ok, I know about Gonorrhoea and this kind of stuff, but I don't know about Chlamydia, and I don't know about this, that and the next thing. And these are all things that children can get, and I need to be able to explain what can happen if they have unprotected sexual relationships' (F2 - Ian, foster carer).* This reduced role overload by providing caregivers with the specialist SHR knowledge that they reported was needed to incorporate SHR discussions into practice. Having more specialised SHR knowledge was also perceived by caregivers as a means of ensuring that LAC were provided with information that was factually correct, not confusing and unable to be misconstrued: *'one worker talked about your flower, and if you needed anything sorted you would go to the flower shop… I don't think that things like that really help when talking about going to clinics and your vulva. You need to use the proper names so that everyone is quite clear as she could have people thinking 'oh right, I need to go buy some flowers'… because they take you literally' (F2 - Laura, residential carer).*

Being able to draw on SHR knowledge improved caregivers' confidence and reduced embarrassment. In turn, this facilitated discussions of SHR with LAC. Caregivers who had received SHR training highlighted the importance of being provided with leaflets and books that they could use to help discuss SHR with LAC: *'after the training I found that I was really more confident and I had all the information on hand and booklets to show' (F3 - Maria, foster carer).* Caregivers also discussed how the training had provided them with details of websites and helplines that they could use if they were asked questions that they did not know the answer to. Having these resources was valued as it allowed caregivers to help answer questions they might not have known the answer to: *'If I've no got an answer for them I'll maybe say 'we've got literature on that so just gie me a minute and we'll go and get it and we'll take 5 min to go through it' (F4 - Patricia, residential carer).* Training resources were particularly valued for helping caregivers tailor SHR discussions to the physical, emotional and cognitive ages of LAC.

While SHR training provided caregivers with the skills needed to incorporate SHR discussions into practice, it should be noted that all of the residential carers and foster carers, regardless of whether they had received SHR training or not, routinely discussed SHR with LAC. The range of topics reported as being discussed within caregivers' narratives included puberty, peer relationships, romantic relationships, masturbation, having sex for the first time, contraception, reproduction and sexual exploitation. The main difference observed was that caregivers who had not received SHR training were more likely to express personal concerns about the accuracy of the information they had provided: *'we were playing Connect 4, and one of the girls said 'how do you get pregnant' and we said 'well, you need to have sex'. 'Aye, I know that… and I know that he cums, but how does that then work?' So we dismantled the Connect 4, and we said 'well it's no square, but you'll have to imagine this is a womb, and these are the fallopian tubes', and we used the wee circles as the sperms and the eggs, and we used that to explain it…. Once we were finished I turned to [another caregiver] and said 'did I get that right?'' (F5 - Claire, residential carer).* These concerns further emphasised the important role that SHR training played in reducing role overload by bolstering caregivers' knowledge and confidence.

### Pastoral support

Being supported by managers to talk to LAC about SHR reduced role overload. The main reason for this being that caregivers felt supported to undertake discussions that they found to be challenging and did not have the skills and training needed to undertake these discussions alone. This was particularly valued when caregivers had to discuss SHR with LAC who had been sexually abused, groomed or exploited and were demonstrating problematic sexual behaviours: *'I found that work really difficult, because I had never had to deal with… trying to manage a child – cos he was a child at the time — who is not only, you know, being abused, but is an abuser… I felt really, you know, unsure of how best to manage that. One of the best things with managing that was that my manager agreed to support me, and we did the work together' (Agnes, social worker).*

Peer networks also reduced role overload by providing caregivers with a group of individuals that they could use as source of help and advice: *'We deal with it pretty well, but I think with this wee core group of carers that we've got there's*

*always an opportunity for learning… 'I've tried to get this boy to do his bloody homework and he just will not do it' and somebody will say 'try this' and you find that it works. That's where our support is… from other carers in our group. We bounce off each other' (G2 - Ian, foster carer).*

## DISCUSSION

Using role theory[40–42] we have identified that the introduction of SHR discussions into caregiving practices for LAC can result in caregivers experiencing role ambiguity, conflict and overload. Providing SHR policies and training, and encouraging caregivers to develop strategies that safeguarded all individuals living within placement were identified as ways in which institutions, in this case social work services, could work to reduce role ambiguity and conflict. Our results also highlight the important role that the institution can play in minimising caregivers' experience of role overload specifically related to the incorporation of SHR discussions into practice, namely through SHR training, the provision of pastoral supports and encouraging the use of interagency working. Contrary to the results of previous studies, which have shown that SHR discussions within LAC settings are limited,[24 27 29] our results demonstrate that caregivers regularly discussed SHR with LAC. These findings further develop existing research that is dominated by atheoretical accounts of the facilitators and barriers of SHR promotion in care settings.

Institutional support for intervention delivery contributes to the successful adoption of general health prevention programmes.[53] Our results support this finding by demonstrating that caregivers incorporate discussing SHR with LAC into their usual caregiving practices when social work departments invest in creating an organisational culture that reduces taboos about sex and clearly outlines what is expected of caregivers. In this study, culture change was underpinned by corporate parenting policy which states that all caregivers have a legal duty to support LAC to obtain the '*same health outcomes… as any good parent would want for their own children*'.[54] This appealed to caregivers' desire to be viewed as being 'good' parents to the children in their care; suggesting that the framing of SHR policies to reflect the professional identity of caregivers can act as an important first step towards the successful incorporation of SHR discussions into practice.

Lack of guidance about who should be responsible for discussing SHR is known to act as a barrier to LAC receiving SHR information and advice in non-LAC settings.[27 28 30 31] Our results show that introducing SHR policies in LAC settings reduced this ambiguity by clarifying role expectations and emphasising that talking to LAC about SHR was the responsibility of all caregivers. This is important as previous studies in non-LAC settings have demonstrated that when role ambiguity occurs, both parents and health professionals choose not to discuss SHR with adolescents due to perceptions that there are others who are better qualified to undertake this role.[55]

While on the whole our results show that policy can be used to clarify the role that caregivers can play in discussing SHR with LAC, our results demonstrate that caregivers required additional guidance on how to confidentially record SHR discussions. Lack of confidentiality is a known barrier to adolescents approaching services for SHR information.[56] Thus, caregivers not being able to record information in a confidential manner has the potential to prohibit LAC from engaging in open SHR discussions with caregivers. To minimise the risk of this happening, we recommend that SHR policies and training should provide clear guidance on how and where caregivers should record SHR discussions, and outline the circumstances under which information sharing is required, namely where child protection concerns exist.[57] This is particularly important in residential settings, where multiple caregivers require access to LAC's health information.

In this study, some of the caregivers interviewed discussed how they had drawn on their personal experiences of discussing SHR with their biological children when undertaking SHR discussions with LAC. While having personal experiences to draw on may facilitate SHR discussions, evidence from secondary schools shows that in the absence of SHR guidance and training, teachers incorporate their own moralistic views of sexual behaviour into SHR discussions with pupils.[58 59] Among our sample, moralistic views were identified as a reason for choosing not to discuss SHR with LAC or opting out of providing information on sensitive issues such as abortion; a finding echoed in other studies looking at healthcare practitioners' attitudes towards contraception and carrying out medical terminations of pregnancy.[47 48] As it is known that adolescents do not seek SHR information from adults they perceive to be judgemental,[60] it is important that caregivers are trained to provide factual, evidence-based information about SHR.

SHR policies and training should take into account the burdens that caring for LAC place on caregivers. Our results highlight that the time demands associated with managing child protection concerns often resulted in SHR discussions not being undertaken unless support from external agencies was sought or the role of meeting the sexual health needs of LAC was undertaken by specialist public health (LAC) nurses. The caregivers interviewed in this study had lots of experience with multiagency working, and their experiences highlight the need for clear delineation of roles when discussing SHR with LAC. To that end, the health plans of LAC should clearly outline: their sexual health needs; what information/intervention is required; and which individuals/agencies will be involved in meeting those needs and how. A lead professional should be identified and tasked with the responsibility of overseeing any multiagency involvement, and ensuring that all individuals/agencies involved have clearly delineated roles and responsibilities.[30] This is particularly important for LAC at risk of sexual exploitation or demonstrating problematic sexual behaviours.

Our results highlighted that caring for LAC who had experienced sexual abuse/exploitation had emotional impacts on caregivers. Caring for young people demonstrating risky behaviours can result in caregivers experiencing vicarious trauma and strain.[61 62] Identifying ways to minimise caregiving strain is important as it is known that this can contribute to placement breakdowns.[61 63] Repeated breakdowns of placements are known to erode the ability of LAC to form secure and trusting relationships with their caregivers,[64] which in turn may lead to LAC being unable to identify a consistent and trusted adult that they can discuss SHR with.[25 28 65 66] Thereby further excluding LAC from accessing SHR information and advice.

Caregiving strain can negatively impact on the sensitivity of caregivers' parenting skills, particularly the ability to respond to the emotional rather than physical ages of children.[61] As the majority of LAC have experienced child maltreatment,[6] known associations between maltreatment, emotional dysregulation and cognitive delay[67] increase the likelihood that caregivers will need to tailor preventative SHR messages to the emotional and cognitive ages of LAC. Working to alleviate caregiving strain is thus an important step to ensuring that age-appropriate SHR discussions are undertaken with LAC. One potential way of alleviating caregiving strain is through peer-led support services. For instance, recent evidence shows that the provision of peer support for parents with children experiencing mental health difficulties reduces isolation, increases social support, empowers caregivers and increases levels of self-care.[68] Future research should explore whether peer-led sexual health support services for caregivers can empower caregivers to undertake SHR discussions, while reducing caregiving strain.

Finally, our results emphasise the importance of training for facilitating SHR discussions within the care setting. SHR training providing caregivers with the confidence, knowledge and skills needed to undertake SHR discussions with LAC. These findings reflect results from other studies of sexual health, which have demonstrated that the introduction of SHR training improves the confidence of teachers and medical professionals to talk with pupils and patients about SHR[69–71] and increases the likelihood of parents engaging in conversations about sex with adolescents.[72] It is encouraging that training facilitated SHR discussions, as evidence shows that adolescents whose parents discuss SHR with them are more likely to delay intercourse, use contraception and have fewer sexual partners.[34–37] Further research should establish whether caregiver-led SHR discussions can contribute to a reduction in sexual risk among LAC.

### Reflections on study conduct

Feminist research practices challenge the view that research can and should be value-free,[73] and advocates that researchers should reflect on how personal characteristics of both the research team and participants may have influenced participation, engagement and the credibility of findings.[74] Here, we briefly reflect on how who we were as individuals and the choices that we made as a research team affected the research process. In doing so we identify some of the methodological limitations and strengths of this study.

Looking first at recruitment and data collection, we are aware that our decision to access caregivers through social work services could result in some caregivers feeling pressured into participating. As we were aware that there was an expectation among senior managers that caregivers 'would' participate we made sure that as researchers we took preventative steps to ensure that caregivers were fully informed of their rights, that is, that they could choose not to participate in the research, could choose how much information they wanted to disclose and that they could withdraw from the study at any point without us informing their employers that they had done so. On the whole, we feel that these steps led to caregivers feeling that they had the right not to participate, with several of those contacted declining to participant. However, one of the caregivers that CN interviewed appeared to be participating out of expectation rather than choice. The data collected from this interview were very closed, lacked the same depth of information that other caregiver interviews contained and was terminated early.

Recruiting caregivers through social work services also raised the issues of power dynamics and bias within the research process. For example, the social work department that we recruited participants through acted as a gatekeeper to participants, and senior managers played an active role in identifying the area-based team that social workers would be recruited from and choosing which residential carers would be contacted. From the perspective of senior management the decisions that they made were designed to minimise any disruptions to service caused by caregivers participating in research; however, it is also possible that these decisions could have resulted in caregivers who were known to be good at discussing SHR with LAC being selected for interview, thus, casting both the social work department and the SHR training that they had introduced in a good light. To counter this possibility, it was agreed that caregivers would be randomly selected from staff lists.

While there was potential for biased narratives to be collected, our analysis of data identified that caregivers were often quite critical of social work services, the training that they had received and the practices of other caregivers. In many ways, the fact that none of the research team had histories of being in care, or providing care to LAC, was beneficial as it meant that we were often viewed as naïve outsiders who needed to be educated about the care system and participant's experiences of caring for LAC. The data generated could have been very different if our interviewer (CN) had had any connections to social work services. For instance, it is possible that the data collected would have been less detailed as a result of participants not fully elaborating on practices, policy and training due to the perception that the researcher already had access to this information.

Finally, it should be noted that our decision to use role theory as a means of interpreting our findings partly reflected the research interests of LE, who was involved in another study exploring how professional role identity shaped the reactions of community nurses to changes in nursing policy. The identification that the themes being coded could be mapped to the definitions of role ambiguity, role conflict and role overload was viewed as a strength by us due to the existing literature relating to the sexual health of LAC being largely atheoretical. Had LE not been familiar with role theory and its application with health settings it is possible that other theoretical lenses would have been used to interpret our research findings; resulting in a potentially different interpretation of caregiver narratives.

### Strengths and limitations

Strengths of this qualitative interview study include the generation of rich qualitative data that provides insight into: (1) How the introduction of SHR training and policies facilitates SHR discussions within the care setting. (2) How caregivers negotiate and address perceived impacts of undertaking SHR discussions on their professional and personal identities. As LAC are often excluded from school-based SHR education, the data generated from this study provide valuable insights into alternative means through which SHR discussions can be undertaken with this population. Role theory underpins the analysis and provides insight beyond existing research that is dominated by atheoretical accounts of the facilitators and barriers of SHR promotion in care settings.

This study has two limitations. First, and most importantly, our study does not capture the experiences of kinship carers or parents whose children may be looked after at home under home supervision orders. Future research should explore how the experiences of these caregivers differ from those in our study who are more professionally oriented, and in particular, the extent to which our understanding of role theory applies. Second, while our aim was to interview caregivers when half of all foster carers, residential carers and social workers had received SHR training, institutional delays in rolling out training to caregivers meant that interviews occurred when half the foster carers, all residential carers and no social workers had received training. While our achieved sample reflects this, it is possible that our findings for social workers in particular do not fully capture the impact of policy that encourages discussion about SHR with LAC. It would be beneficial if future revisions of sexual health training and policy within the care system reflected evaluative work being undertaken in the field of nursing[75] and collected prequantitative and postquantitative as well as prequalitative and postqualitative data on the effect that introducing new policy has on the constructions of caregivers' professional identity.

## CONCLUSION

To encourage SHR discussions within the care setting it is important that organisations adopt a culture that is supportive of LAC receiving SHR education. While the provision of sexual health policies and training are important first steps towards caregivers' discussing SHR with LAC, both institutional factors and caregivers' own personal experiences and values can act as barriers to this occurring. In order for SHR discussions to become a routine aspect of caregiving, our results suggest that future revisions of SHR policy and training by social work departments should emphasise to caregivers the importance of collaborative working, the use of safe care procedures and the availability of managerial and pastoral support to help caregivers overcome these barriers and include SHR discussions within their practice. Emphasis also needs to be placed on ensuring that inclusion of SHR within caregiving practices does not increase levels of caregiver strain or erode caregivers' sense of professional identity by creating role confusion. Further research is needed to ascertain whether caregiver-led SHR discussions improve caregivers' professional identity and reduce the sexual vulnerability of LAC.

**Acknowledgements** The authors thank the Local Authority for their participation in this study, and all of the individuals who took part in the study and shared their thoughts and experiences. The authors also thank Professor Lisa McDaid, Professor Daniel Wight and Dr Alice Maclachlan for commenting upon this manuscript.

**Contributors** CN designed the study, carried out the qualitative data collection and analysis, and drafted the manuscript. MH and LE participated in the design of the study and drafting of the manuscript. All authors read and approved the final manuscript.

**Funding** This work was supported by the UK Medical Research Council, the Scottish Government Chief Scientist Office, Edinburgh Napier University and Glasgow Caledonian University. CN was part of the Children, Young People, Families and Health and Social Relationships and Health Improvement Programmes [MC_UU_12017/2, MC_UU_12017/11 and SPHSU11], and MH was part of the Children, Young People, Families and Health, Social Relationships and Health Improvement, and Understanding and Improving Health within Settings and Organisations Programmes [MC_UU_12017/2, MC_UU_12017/11, MC_UU_12017/12 and SPHSU11] all at the MRC/CSO Social and Public Health Sciences Unit, University of Glasgow. CN was funded by a Scottish Government Chief Scientist Office Pre-doctoral Fellowship (CSO DTF/09/16). LE was core funded by the Department of Nursing and Community Health, Glasgow Caledonian University. The funders had no role in study design, data collection and analysis, decision to publish, or preparation of the manuscript.

**Competing interests** None declared.

**Patient consent for publication** Not required.

**Ethics approval** The College of Social Science Research Ethics Committee at the University of Glasgow reviewed and approved the study and consent procedure (Application number: SS/10/0027).

**Provenance and peer review** Not commissioned; externally peer reviewed.

**Data sharing statement** No further data are available as this was not part of the ethics approval.

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
