## [Reviewer comments · BMJ Open]

ARTICLE DETAILS

TITLE (PROVISIONAL)	Providing sex and relationships education for looked after children: a qualitative exploration of how personal and institutional factors promote or limit the experience of role ambiguity conflict and overload among caregivers
AUTHORS	Nixon, Catherine; Elliott, Lawrie; Henderson, Marion

VERSION 1 - REVIEW

REVIEWER	Suzanne Dawson Flinders University Adelaide, Australia
REVIEW RETURNED	30-Jul-2018

GENERAL COMMENTS	Thank you for the opportunity to review this paper. Overall the paper is well written and contributes to the discussion on this important topic. I would suggest the authors consider making a few minor changes to strengthen the paper in two areas. Methods Consider providing some clarity as to why the aim was to conduct the study after half of all participants had received SHR training, especially given this difference was not distinguished in the results. Additionally more information could be provided if available on the participants (age, gender, and experience in working with LAC) as well as some descriptive details about each participant group for comparison for international readers (e.g. typical numbers of children in residential group homes, case-load numbers for SWs). Finally, the authors could consider adding a sample of the interview schedule or entire schedule in the appendices. Results Whilst this was well written, some re-working of this section could aid the reader in following the information presented and provide more clarity. For example, the Table could better link to the headings and sub-headings in this section. Additionally short summaries could be added under each heading to explain why the various sub-headings were grouped together. Given the objective was to distinguish between institutional and personal factors promoting or limiting the role of caregivers in promoting SHR among LAC, more clarity and a summary of this would also be helpful. This could be done in the Table as well as a brief written summary.
--

REVIEWER	Dionne Gesink University of Toronto, Canada
REVIEW RETURNED	29-Dec-2018

GENERAL COMMENTS	This is a thoughtfully and sensitively written manuscript presenting the results of a study exploring how social services care providers perceive their role in educating children in care on sexual health and relationships. The introduction is well written and the results are important. Additional details are needed in the methods and results sections. There is some redundancy, primarily in the discussion (i.e. representing results in detail), that can be removed to address my comments. Introduction:  Two research questions are provided. Only one is needed. The manuscript that follows is more aligned with the second research question presented. The first question is not answered. Please revise. Methods:  Please include the qualitative study design (e.g. grounded theory, phenomenology, thematic analysis, framework analysis...). In-depth interviews are the data collection method. Please include where interviews were conducted, by who, how long interviews took, and if participants were compensated for their time/knowledge. Please clarify if transcripts were verified before coding. Please provide a sense of who was involved in coding. Please explain why role theory was not used to inform the coding framework during analysis. Currently, it seems role theory was limited to interpreting the results? Page 7, lines 132-133: "A selection of transcripts and the coding framework were then reviewed by independent reviewers...". Please clarify what the transcripts and coding framework were reviewed for. Page 7: "Reflections upon how researcher characteristics may have influenced the research are reported elsewhere [50]." Please provide a brief reflection so readers have enough information to evaluate this manuscript without having to go to another. Please explain how the final sample size was arrived at and whether saturation was achieved. Results:  Page 9, "Role Conflict": Please provide evidence, in the form of illustrative quotes or thick description, to support your interpretation and strengthen the section. Page 10, lines 227-232: "Monitoring the sexual health of LAC acted as a source of conflict when caregivers had to balance the SHR needs of LAC against protecting the safety, wellbeing and livelihoods of other household and family members. For example, several FCs discussed how allegations of improper behaviour could have financial/reputational repercussions, and affect the other roles they fulfilled: "it's worrying... my son's a police officer, my husband works in law enforcement and I work with students – we have to take extra care". This passage is unclear and confusing. At first it sounds like the authors are talking about related LAC, then coworkers, which then makes the passage sound like workers are covering up professional misconduct. It's not until after the quote that it becomes more clear that the authors
--

are talking about the relatives of workers who do not work with LAC but work in other social services. The quote only clarifies who the subject of the passage is, but does not provide evidence of how talking about SHR with LAC threatens workers families.

Please strengthen and clarify this section.

12. Page 11, lines 233-235: "Caregivers' concerns affected how SHR discussions were undertaken, with all of those interviewed emphasizing the importance of using 'safe care' practices to minimize the risk of harm to caregivers." Please unpack this sentence - what is meant by "caregiver concerns", "safe care' practices", and "risk of harm to caregivers".

13. Page 12, "Workload": Please provide evidence, in the form of illustrative quotes or thick description, to support your interpretation.

14. Page 12, lines 286-288: "Collaborating with external agencies was identified as reducing role overload and ensuring that LAC received help and support: "we're limited wi' [with] staff... so it's a bit about [about] sharing you know? We kind of all come together" (Patricia, RC)." The associated quote is weak and does not provide evidence to support the interpretation without further explanation. Please correct.

15. Consider adding illustrative quotes to Table 1 to strengthen the results.

Discussion:

16. Page 21, lines 528-530: "First, this was an exploratory study with a relatively small sample of caregivers recruited from a single local authority, therefore generalizing the findings from this study to the practices of other caregivers may not be possible."

Generalizability is an important quantitative methodologic concept, but is neither the goal of qualitative methodology nor a limitation. It is not appropriate to evaluate the rigor of a qualitative study using generalizability. Transferability can be discussed in the context of qualitative methods and rigor, but not generalizability. Additionally, small sample sizes are appropriate for qualitative studies, especially if saturation is achieved. Please revise.

17. Page 21, lines 541-543: "The final limitation is that while we have used role theory as a theoretical framework for interpreting our findings..." Using theory in a qualitative study is not a limitation. This is a bit like saying a quantitative study was limited by not presenting all the possible associations available in a quantitative dataset. Please revise.

18. The discussion can be strengthened by adding reflexivity. How did who the researchers are influence (both help and hinder) the study, thinking through recruitment, data collection, data analysis and interpretation of findings? Reflexivity also helps assess the rigor and validity of qualitative studies.

Minor comments:

19. There is excessive use of acronyms throughout this manuscript. Several of these acronyms are not defined in full, and several are unintuitive and place unnecessary burden on the reader (for example FC, RC, SW, CP, SRQR). Two examples (among many others) of the difficulty overuse of acronyms present include (page 9): "RCs and FCs experienced challenges balancing the competing demands of CP concerns with undertaking preventative SHR interactions" and (page 13): "Lack of time, combined with perceptions that LAC and their families distrusted SWs, resulted in SWs reporting that RCs and FCs were better

	placed to discuss SHR with LAC". Please reduce the number of acronyms to improve readability. 20. Page 4, lines 62-63: "The poor sexual health of LAC may be exacerbated by limited access to sexual health and relationships (SHR) education." To reduce stigma and blanket judgement, consider rewording to, "The sexual health vulnerability of LAC may be exacerbated by limited access to sexual health and relationships (SHR) education." 21. Throughout – several paragraphs contain more than one thought and can afford to be broken into new paragraphs.
--	--

VERSION 1 – AUTHOR RESPONSE

Response to Reviewer: 1

=====

METHODS

1. Consider providing some clarity as to why the aim was to conduct the study after half of all participants had received SHR training, especially given this difference was not distinguished in the results.

To address this comment we have amended the text to read: "Our aim was to interview caregivers when half of all foster carers, residential carers and social workers had received training as it was felt that doing so would more accurately capture the role that training had upon the inclusion of SHR discussions for each of the different caregiving roles". The revised text can be viewed on lines 150-152 of the manuscript.

2. Additionally more information could be provided if available on the participants (age, gender, and experience in working with LAC) as well as some descriptive details about each participant group for comparison for international readers (e.g. typical numbers of children in residential group homes, case-load numbers for SWs).

Information about participants age and gender have been added into the methods section, along with details about current caseloads and the number of children looked after in placement. These changes can be viewed on lines 159-166 of the manuscript.

3. Finally, the authors could consider adding a sample of the interview schedule or entire schedule in the appendices.

The interview schedule has been upload and is titled Supplementary File 1

RESULTS

1. Whilst this was well written, some re-working of this section could aid the reader in following the information presented and provide more clarity. For example, the Table could better link to the headings and sub-headings in this section. Additionally short summaries could be added under each heading to explain why the various sub-headings were grouped together.

To address this comment we have amended each of the headings in the main body of results to better reflect those provided in Table 1. At the start of the result sections for role ambiguity, role conflict and role overload we have included definitions of those concepts to remind the reader of what these entail. How each of the themes identified within the sub-heading links to this is then highlighted.

For instance, on line 245 we have added the following text to identify that the process of clarifying role expectations reduced role overload among caregivers: "Among residential carers and social workers, incorporating institutional expectations about corporate parenting responsibilities into practice resulted in their developing personally held views about what it meant to be a "good" corporate parent". Each of these changes has been noted within the marked copy of the manuscript.

2. Given the objective was to distinguish between institutional and personal factors promoting or limiting the role of caregivers in promoting SHR among LAC, more clarity and a summary of this would also be helpful. This could be done in the Table as well as a brief written summary.

To address both this comment and the recommendation by Reviewer 2 that illustrative quotes be added to Table 1 we have restructured the table to have 4 columns. Column 1 identifies the themes and associated quotes, column 2 identifies whether the theme operated at an institutional or personal level, column 3 identifies how the theme acted as a barrier or facilitator of SHR discussions and column 4 identified whether the theme is an indicator of role ambiguity, conflict or overload. Throughout the results section we have added text to flag to the reader how the themes being discussed interact to influence the likelihood of SHR discussions being incorporated into practice. Please see the section on balancing competing demands of child protection and preventative SHR work (line 299) for an example of how this has been done. Each of these changes has been noted within the marked copy of the manuscript.

=====

Response to Reviewer 2

=====

To address the comment that there is some redundancy in the discussion section, caused by representing results in detail, we have restructured the discussion by removing all text that provided results without reference to the existing literature or recommendations for practice. These changes have resulted in the discussion being shortened and restructured. All changes are noted within the marked copy of the manuscript.

INTRODUCTION

1. Two research questions are provided. Only one is needed. The manuscript that follows is more aligned with the second research question presented. The first question is not answered. Please revise.

We have removed the text "we explore how caregivers of looked after young people construe their roles as promoters of sexual health" as we believe that this was inadvertently making it seem that two questions were being posed instead of one. This amendment can be viewed on lines 125-131.

METHODS

2. Please include the qualitative study design (e.g. grounded theory, phenomenology, thematic analysis, framework analysis...). In-depth interviews are the data collection method.

We have revised the text that stated data were analysed thematically to read 'thematic analysis was used to analyse the data generated' to emphasise the qualitative study design to the reader. These changes can be found on line 191.

3. Please include where interviews were conducted, by who, how long interviews took, and if participants were compensated for their time/knowledge.

This information has been added into the manuscript on lines 175-180.

4. Please clarify if transcripts were verified before coding.

Information on the processes used to verify the qualitative transcripts has been added into the manuscript on lines 184-188.

5. Please provide a sense of who was involved in coding.

The methods section has been amended to provide information outlining who was involved in coding and what role they played at each stage. These changes can be found on lines 191-209.

6. Please explain why role theory was not used to inform the coding framework during analysis. Currently, it seems role theory was limited to interpreting the results?

We have clarified the text within the methods section to highlight to the reader that role theory was used as both an analytical and interpretative tool. The text relating to this can be found on lines 196-203.

7. Page 7, lines 132-133: "A selection of transcripts and the coding framework were then reviewed by independent reviewers...". Please clarify what the transcripts and coding framework were reviewed for.

This information has been added into the manuscript on lines 195-197.

8. Page 7: "Reflections upon how researcher characteristics may have influenced the research are reported elsewhere [50]." Please provide a brief reflection so readers have enough information to evaluate this manuscript without having to go to another.

This text has been removed. Reflections upon how research characteristics may have influenced the research have been included within the discussion in a section focussed upon reflexivity. This section can be found on pages 26-27 of the manuscript.

9. Please explain how the final sample size was arrived at and whether saturation was achieved.

Additional information about sampling has been included in the manuscript on lines 142-148. These outline the constraints that were placed upon the research team by social work services in order to minimise service disruption. The final sample size reflects the number of the individuals who were approached from the lists generated by social work services who agreed to participate. We have included information about how data saturation was assessed on lines 207-209 of the manuscript.

RESULTS

10. Page 9, "Role Conflict": Please provide evidence, in the form of illustrative quotes or thick description, to support your interpretation and strengthen the section.

The text that was previously provided here was an overview of the topics that would be discussed within the subheadings. As such, we feel that providing illustrative quotes to strengthen this section would result in duplication. To make it more obvious to the reader that this is an overview we have removed the text that reads: "Factors that reduced role conflict and facilitated SHR discussions included: developing strategies to monitor LAC's sexual behaviours and protect against the risk of allegations being made; caregivers having experiences of discussing SHR with their biological children to draw upon and help frame discussions with LAC; and caregivers reflecting upon the school- and home-based SHR education that they had received as motivation for undertaking SHR discussions. These themes, which will now be discussed in more detail, were predominantly raised by residential carers and foster carers". And, replaced it with the sentence: "These challenges, and the approaches that caregivers used to address them, will now be discussed". These changes can be found on lines: 287-297 of the manuscript.

11. Page 10, lines 227-232: "Monitoring the sexual health of LAC acted as a source of conflict when caregivers had to balance the SHR needs of LAC against protecting the safety, wellbeing and livelihoods of other household and family members. For example, several FCs discussed how allegations of improper behaviour could have financial/reputational repercussions, and affect the other roles they fulfilled: "it's worrying... my son's a police officer, my husband works in law enforcement and I work with students –we have to take extra care". This passage is unclear and confusing. At first it sounds like the authors are talking about related LAC, then coworkers, which then makes the passage sound like workers are covering up professional misconduct. It's not until after the quote that it becomes more clear that the authors are talking about the relatives of workers who do not work with LAC but work in other social services. The quote only clarifies who the subject of the passage is, but does not provide evidence of how talking about SHR with LAC threatens workers families. Please strengthen and clarify this section.

To address these comments we have redrafted this section completely. In doing so we have emphasised the vulnerability that caregivers felt when undertaking SHR discussions with LAC and the concerns that they had about young people making false accusations. These changes can be found on lines 341-361 of the manuscript.

12. Page 11, lines 233-235: "Caregivers' concerns affected how SHR discussions were undertaken, with all of those interviewed emphasizing the importance of using 'safe care' practices to minimize the risk of harm to caregivers." Please unpack this sentence - what is meant by "caregiver concerns", "safe care' practices", and "risk of harm to caregivers".

We have clarified that 'risk of harm' referred to the threat of false allegations being made against caregivers. We have also added illustrative quotes into Table 1 to provide examples of the safe care practices used by caregivers. By doing this we hope to provide the reader with greater clarity of the processes that caregivers use to keep themselves and other members of the household safe. These changes can be viewed on lines 363-371.

13. Page 12, "Workload": Please provide evidence, in the form of illustrative quotes or thick description, to support your interpretation.

The text that was previously provided here was an overview of the topics that would be discussed within the subheadings. As such, we feel that providing illustrative quotes to strengthen this section would result in duplication. To make it more obvious to the reader that this is an overview we have removed the text related to the factors that reduced the experience of role overload and replaced it with the following sentence: "These barriers, along with the strategies used to address these, will now be discussed". These changes can be viewed on lines 411-419 of the manuscript.

14. Page 12, lines 286-288: "Collaborating with external agencies was identified as reducing role overload and ensuring that LAC received help and support: "we're limited w/ [with] staff... so it's a bit about [about] sharing you know? We kind of all come together" (Patricia, RC)." The associated quote is weak and does not provide evidence to support the interpretation without further explanation. Please correct.

We have extended and restructured the section on workforce capacity and provided additional illustrative quotes within Table 1 to highlight how caregivers utilised external agencies to support caregivers in monitoring behaviour and undertaking SHR discussions. The changes specifically relating to this example can be found on lines 423-433 and 455-462. The wider restructuring can be seen across the 'workforce capacity' subheading.

15. Consider adding illustrative quotes to Table 1 to strengthen the results.

Illustrative quotes have been added to Table 1, which has also been restructured per the suggestion of Reviewer 1.

DISCUSSION

16. Page 21, lines 528-530: "First, this was an exploratory study with a relatively small sample of caregivers recruited from a single local authority, therefore generalizing the findings from this study to the practices of other caregivers may not be possible." Generalizability is an important quantitative methodologic concept, but is neither the goal of qualitative methodology nor a limitation. It is not appropriate to evaluate the rigor of a qualitative study using generalizability. Transferability can be discussed in the context of qualitative methods and rigor, but not generalizability. Additionally, small sample sizes are appropriate for qualitative studies, especially if saturation is achieved. Please revise.

We have removed all references to generalisaility and instead restructured the limitations section to focus on identifying that additional research is needed to explore 1) how the experiences of kinship carers and parents whose children are looked after under home supervision orders differ from those reported by the caregivers we interviewed the voices not captured within the study and the implications and 2) whether our understandings of role theory apply to different caregiving contexts. These changes can be found on lines 756-770.

17. Page 21, lines 541-543: "The final limitation is that while we have used role theory as a theoretical framework for interpreting our findings..." Using theory in a qualitative study is not a limitation. This is a bit like saying a quantitative study was limited by not presenting all the possible associations available in a quantitative dataset. Please revise.

We have removed all references to the application of role theory to this setting from the limitations section and instead indicated that by underpinning our analysis with theory we have further developed research in a field that is largely atheoretical. These changes can be viewed on lines 751-754. We have also flagged how future resarch within this field could benefit from the application of role theory on lines 767-770.

18. The discussion can be strengthened by adding reflexivity. How did who the researchers are influence (both help and hinder) the study, thinking through recruitment, data collection, data analysis and interpretation of findings? Reflexivity also helps assess the rigor and validity of qualitative studies.

A section on reflexivity has been added into the discussion on pages 26-27. To tie this in with the section on strengths and limitations we have focussed upon exploring how researcher characteristics and the power differentials that arise when recruiting through gatekeepers could have biased the sample, the narratives generated and our analysis.

19. There is excessive use of acronyms throughout this manuscript. Several of these acronyms are not defined in full, and several are unintuitive and place unnecessary burden on the reader (for example FC, RC, SW, CP, SRQR). Two examples (among many others) of the difficulty overuse of acronyms present include (page 9): "RCs and FCs experienced challenges balancing the competing demands of CP concerns with undertaking preventative SHR interactions" and (page 13): "Lack of time, combined with perceptions that LAC and their families distrusted SWs, resulted in SWs reporting that RCs and FCs were better placed to discuss SHR with LAC". Please reduce the number of acronyms to improve readability.

In response to this point we have limited the use of acronyms to SHR (sexual health and relationships) and LAC (looked-after child). All other acronyms have been removed and replaced with full text.

20. Page 4, lines 62-63: "The poor sexual health of LAC may be exacerbated by limited access to sexual health and relationships (SHR) education." To reduce stigma and blanket judgement, consider rewording to, "The sexual health vulnerability of LAC may be exacerbated by limited access to sexual health and relationships (SHR) education."

We have edited the text per your suggestion. The revised text can be found on line 79.

21. Throughout – several paragraphs contain more than one thought and can afford to be broken into new paragraphs.

We have broken a number of paragraphs across the document into new paragraphs to increase readability. Instances of this are indicated in the marked up manuscript.

VERSION 2 – REVIEW

REVIEWER	Dionne Gesink University of Toronto, Canada
REVIEW RETURNED	21-Feb-2019

GENERAL COMMENTS	The authors have been thorough responding to my comments and concerns. I find the manuscript much stronger, more clear, and easier to follow. The results are compelling. I have no additional concerns or comments.
--